

# The association between mental rotation capacity and motor impairment in children with obesity—an exploratory study

Frederik J.A. Deconinck[1,*], Eva D'Hondt[2], Karen Caeyenberghs[3], Matthieu Lenoir[1] and Mireille J.C.M. Augustijn[1,*]

[1] Department of Movement and Sports Sciences, Ghent University, Ghent, Belgium
[2] Faculty of Physical Education and Physiotherapy, Vrije Universiteit Brussel, Brussels, Belgium
[3] Cognitive Neurosciences Unit, Deakin University, Melbourne, Victoria, Australia
[*] These authors contributed equally to this work.

## ABSTRACT

**Background**. Motor impairments are relatively common in children with obesity and evidence suggests that these difficulties go beyond those expected based on the extra weight. This study aimed to investigate the mental rotation capacity in children with obesity, i.e., the ability to mentally view and rotate spatial information, which is a function of both visual-spatial and action representation processes. In particular, we examined whether children with obesity solved mental hand rotation tasks using an egocentric perspective (i.e., motor imagery) and if performance was related to their motor competence.

**Methods**. Fifty children (age range: 7–11 y) of which 19 with obesity and motor impairments (OB-) as assessed by the Movement Assessment Battery for Children (2nd version), 13 with obesity without motor impairments (OB+) and 18 control children with a healthy weight (HW) and normal motor competence, were submitted to a classic hand rotation task. Sitting at a desk the children were instructed to indicate the laterality of a picture of a hand displayed on a monitor as quickly and as accurately as possible.

**Results**. The results indicate no differences in response time between groups (2,648 ms, 2,558 ms, 2,664 ms for OB-, OB+ and HW respectively). The OB- group, however, had significantly lower accuracy rates and inverse efficiency scores than the HW group (Accuracy: 72% vs. 89%; Inverse efficiency: 4,428 vs. 3,238). No difference was observed in accuracy and inverse efficiency between the OB+ and HW group (Accuracy: 86%; Inverse efficiency: 3,432). In all groups, slower and more error-prone responses were observed when the angle of rotation was larger and when the hand on display was incongruent with the posture of the participants, which indicates that judgments were made from an egocentric perspective and involved motor imagery.

**Conclusion**. All children, including those with obesity, appear to engage in motor imagery. This notion needs to be investigated further in children with obesity and motor impairments, given their generally lower accuracy and decreased efficiency, which may indicate a reduced mental rotation capacity.

Corresponding author
Frederik J.A. Deconinck,
frederik.deconinck@ugent.be

## INTRODUCTION

While recent figures indicate that the increase in body mass index (BMI) in children and adolescents has plateaued in high-income countries, the level is still at an all-time high. Moreover, trends still are accelerating in other parts of the world (e.g., in Asia), which makes pediatric obesity arguably one of today's largest health challenges (*Abarca-Gómez et al., 2017*; *Ng et al., 2014*; *Lobstein et al., 2015*). There is conclusive evidence that children with obesity are more likely to stay obese during adolescence and adulthood, which is associated with an increased risk for non-communicable diseases such as cardiovascular diseases, diabetes, musculoskeletal problems and certain types of cancer (*Simmonds et al., 2016*). The focus of the current paper is on the motor competence of children with obesity, a factor that has received increased attention over the past years (e.g., *Robinson et al., 2015*; *D'Hondt et al., 2009*; *Augustijn et al., 2018a*). Based on a number of studies, the importance of motor competence in the development of a healthy and active lifestyle is indisputable (see *Robinson et al., 2015* for a review).

There is now compelling evidence that the general motor competence of children with obesity is significantly below the level of their peers. On average children with obesity have lower scores on gross and fine motor skills as well as balance (e.g., *Gentier et al., 2013b*; *D'Hondt et al., 2011a*; *Petrolini, Iughetti & Bernasconi, 1995*; *Deforche et al., 2009*). This is perhaps not surprising, because excess (fat) mass leads to greater inertial forces, which are harder to control and coordinate and therefore hamper movement. Indeed, we found that weight loss after a multidisciplinary treatment is accompanied with an increase in motor competence and may explain up to around 25% of the improvement in gross motor skills (*D'Hondt et al., 2011b*; *Augustijn et al., 2018c*). It is important to note, however, that the motor impairment found in children with obesity cannot solely be attributed to the presence of excess (fat) mass. In fact, according to our previous work between 50 and 70% of children with obesity demonstrate levels of motor competence below the 5th percentile, which would indicate a motor impairment that would be associated with more fundamental motor control problems (*D'Hondt et al., 2009*; *Augustijn et al., 2018a*; *D'Hondt et al., 2011a*). For example, even in reaction time and eye-hand coordination tracking tasks children with obesity perform slower and less accurate than healthy weight control children (*Gentier et al., 2013b*; *Petrolini, Iughetti & Bernasconi, 1995*; *D'Hondt et al., 2008*; *Gentier et al., 2013a*). These fine motor tasks require only small movements of arm and hand, which makes the interference of excessive mass negligible. Instead, difficulties during the execution of such tasks suggest deficient central processes related to the perception, planning and control of motor actions similar to those found in children with mild motor impairments, such as developmental coordination disorder (DCD) (*Wilson et al., 2013*).

Although the body of evidence for these more fundamental motor problems in individuals with obesity is currently limited, the findings are compelling enough to warrant further consideration. Motor problems of this kind may pose an extra threat to the individual's health, given the relationship with physical activity, fitness, and mental disorders (*Faught et al., 2005*; *Cairney et al., 2019*; *Gagnon-Roy, Jasmin & Camden, 2016*), and therefore require specific attention. In order to explore the notion of fundamental motor problems in this population further, the current study investigates one of the processes underlying action planning and control, i.e., mental rotation.

Mental rotation can be defined as the ability to mentally view a representation of spatial information and to transform this representation through rotation (*Shephard & Metzler, 1971*). This ability, which involves both visual-spatial and action representation processes, is typically tested by asking a subject to judge whether a 2D or 3D geometric shape on display is identical to a reference shape that may have a different orientation (see Fig. 1). Chronometric studies have demonstrated that the response time of this judgment increases with increasing angular disparity between the stimulus on display and the reference (*Shephard & Metzler, 1971*). Moreover, neuroimaging studies have shown that mental rotation engages motor areas, such as the premotor and supplementary motor area (*Richter et al., 2000*), especially in a hand laterality judgement task, where the participant is required to indicate the laterality of a normal of mirror-reversed of a hand (HLT; see Fig. 1 (*Kosslyn et al., 1998*; *Kosslyn et al., 2001*; *Parsons et al., 1995*; *Parsons, 1987*)). In this case it is thought that the subject uses an egocentric (1st person) perspective and solves the task by imagining moving his/her own hand into the position of the stimulus, also known as motor imagery. Interestingly, motor imagery is embodied and obeys the same rules as actual movements, which implies that the response times in a motor imagery task such as the HLT, are longer for laterally rotated stimuli than for medially rotated stimuli (*Parsons, 1987*). The explanation is that a lateral hand rotation is more complex than a medial rotation due to the anatomical structure and biomechanical constraints (*Parsons, 1987*; *Funk, Brugger & Wilkening, 2005*; *De Lange, Helmich & Toni, 2006*). Furthermore, response times are longer when the posture of the subject is incongruent with the posture of the stimulus, e.g., participants hold hands with palms down while the hand on display is faced with palms up. Mental rotation tasks may also be solved from a 3rd person or object-based perspective. This type of mental rotation uses a non-embodied approach, meaning that responses do not obey the anatomical and biomechanical constraints that act on the body, while there is a linear relationship between the response times and angular disparity between the stimuli (*Shephard & Metzler, 1971*; *Funk, Brugger & Wilkening, 2005*; *De Lange, Helmich & Toni, 2006*).

The ability to mentally manipulate body or object-related information develops through childhood. Although there seems to be considerable interindividual variation, the increase in performance may start from 3 years onwards and is a function of the degree of familiarity with the stimulus rotation, processing time and spatial memory (*Frick, Hansen & Newcombe, 2013*; *Kail, 1997*; *Kail, Pelligrino & Carter, 1980*). Given the link with spatial memory, it is evident that mental rotation ability is related to problem solving (*Geary et al., 2000*) and the acquisition of mathematical knowledge

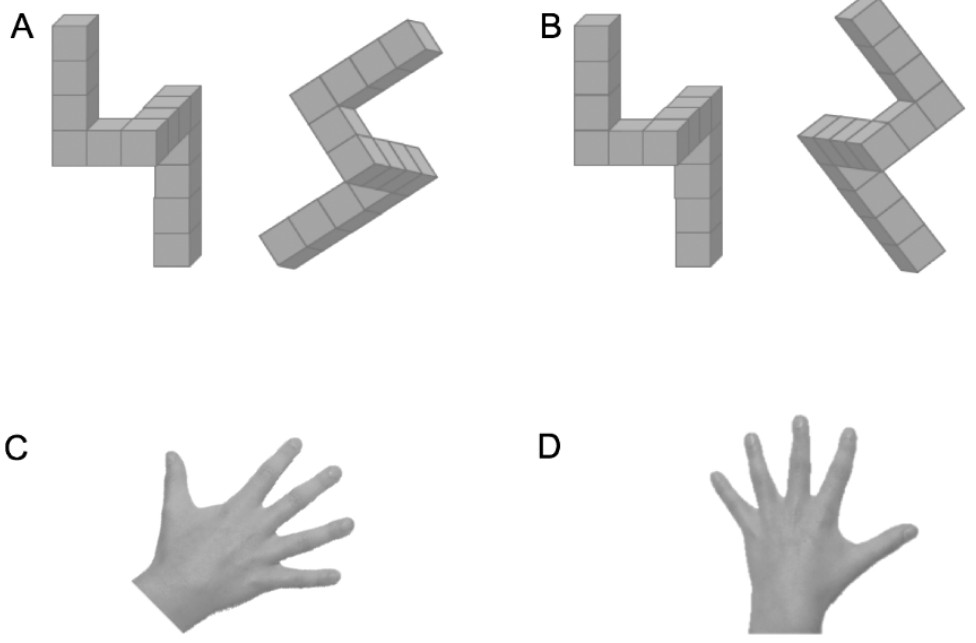

**Figure 1** **Four typical stimuli for a mental rotation task.** In A and B the task is to judge whether the 3D shapes on the right is identical to or a mirror reversed image of the reference shape on the left. In C and D the task to judge whether the hand is a left or a right hand.

(*Hegarty & Kozhevnikov, 1999*). The special case of solving mental rotation tasks using an egocentric perspective involving motor imagery increases through childhood, between the ages of 5 and 12 and as a function of an individual's internal action representation capacity (*Funk, Brugger & Wilkening, 2005*; *Caeyenberghs et al., 2009*; *Spruijt et al., 2015*). Moreover, accumulating evidence indicates that motor imagery is related to actual motor competence. For example, Kaltner and colleagues have recently demonstrated that very active individuals (five training sessions per week on average) outperform their less active peers (one training session per week on average) on a task evoking egocentric mental rotation (*Kaltner, Riecke & Jansen, 2014*). Other research shows that individuals with DCD, a neurodevelopmental disorder characterized by difficulties with the acquisition and execution of motor skills, consistently show slower and/or decreased accuracy on the hand laterality task (*Adams et al., 2017*; *Adams et al., 2014*; *Deconinck et al., 2009*; *Hyde et al., 2014*; *Fuelscher et al., 2015*; *Reynolds et al., 2015*; *Williams et al., 2006*). Some researchers have also found atypical response profiles that reflect the use of an 3rd person perspective (*Wilson et al., 2004*). The finding that the performance in the hand laterality task is impaired in individuals with DCD, is suggested to indicate compromised internal representations in this population, which would explain difficulties with predictive control (*Wolpert, Diedrichsen & Flanagan, 2011*; *Desmurget & Grafton, 2000*).

*Jansen et al. (2011)* found that the mental rotation capacity of 10-year-old children who are overweight was impaired in a typical letter rotation task. In this task two letters are presented alongside each other and the participant is instructed to indicated whether the

letter on the right is the same or a mirror-reversed image of the one on the left. While the response time and the performance profile were similar to that of their healthy weight counterparts, children with overweight had significantly larger error rates, especially in the more complex trials (i.e., when the angular disparity was larger). Given that only non-body related stimuli (i.e., letters) were used, the findings of this mental rotation task suggest compromised visual-spatial function, consistent with previous observations in this population (*Li et al., 2008*; *Augustijn et al., 2018b*). Interestingly, though, a regression analysis indicated that almost 30% of the variance in mental rotation performance was explained by children's motor competence. In a second study on mental rotation in individuals with obesity, this time in adolescents, *Kaltner, Schulz & Jansen (2017)* contrasted stimuli that elicited either an egocentric or object-based strategy. The adolescents with obesity were found to be slower and less accurate than their counterparts with a healthy weight in both conditions. There appeared to be no indication that egocentric or object-based rotation would be more impaired, however, the finding that individuals with obesity were affected more by the angular disparity seemed to suggest a more generalized difficulty with mental rotation. The design of this study was relatively simple and did not check the influence of hand posture or direction of rotation of the stimuli, which are known manipulations to test whether the response involves an embodied perspective, and thus, an internal action representation (*De Lange, Helmich & Toni, 2006*).

The current study set out to investigate the notion of impaired mental rotation in children with obesity further. More specifically, the aim was to investigate the role of motor processes in mental rotation in children with obesity by comparing their performance with typically developing peers and be checking the influence of anatomical and biomechanical constraints on motor imagery. Based on previous research, slower, less accurate performance, or reduced influence of anatomical or biomechanical constraints were considered to indicate impaired motor imagery ability. A second aim was to explore to what extent potential mental rotation difficulties in children with obesity were related with the reduced motor competence found in this population. To this end, the performance on the mental rotation task of children with obesity and co-occurring motor impairments was contrasted with that of children with obesity without clear cut motor impairments.

## MATERIALS & METHODS

### Participants

For this study we initially recruited fifty-seven children, aged between 7 and 11 years. Thirty-two participants were children with obesity (14 boys, 18 girls; mean age $= 9.6 \pm 1.1$), recruited from a specialized rehabilitation center at the start of the actual treatment. Obesity was determined using the international cut-off points standardized for age published by *Cole & Lobstein (2012)*. Five children of this group (2 boys, 3 girls) attended a school for special education. These children were free from severe neurological conditions, but required individualized education due to a delay at school. The healthy weight control children ($N = 25$, 18 boys, 7 girls, mean age $= 9.5 \pm 1.6$) were randomly selected from a local database of regular primary school children considering the age range of the group with

obesity (range: +/− 6 months). The parents of all children gave written informed consent prior to data collection and the protocol of the study was approved by the Ethics Committee of the Faculty of Medicine and Health Sciences of Ghent University (2014/0003).

To identify children with a motor impairment, we used the Movement Assessment Battery for Children, 2nd version (MABC-2) (*Henderson, Sugden & Barnett, 2007*; *Smits-Engelsman et al., 2010*). The MABC-2 consists of 8 items, clustered into three domains (i.e., manual dexterity, ball skills and balance) and has good reliability and validity (*Cools et al., 2009*). Using the available norms for Dutch children, a standard score and a percentile score, both for the total general motor competence score and per cluster, were calculated for each child. The general motor competence scores indicated that 19 out of 32 children with obesity had a general motor impairment, as indicated by a motor competence score at or below the 5th percentile. This group was labeled OB-. Of the other children with obesity (OB+; $N = 13$), six scored above the 16th percentile and seven had scores between percentile 6 and 16, which would indicate being at risk of a motor impairment. Of note, the five children attending a school for special education were all part of the OB- group. In the group with children with HW, 2 children had a general motor competence score at or below the 5th percentile, and 5 scored at or below the 16th percentile. As we wanted to compare the performance of the children with OB against a group without motor impairments we excluded these children from the study, which resulted in sample of 18 children with HW (see Table 1 for an overview of the sample and the division into three groups).

## Materials and procedure

Children's body height (0.1 cm) was measured barefoot using a calibrated stadiometer (Harpenden, Holtain Ltd., Crymych, UK). Additionally, body mass (0.1 kg) and percentage body fat (0.1%) were obtained by means of a digital balance scale with bioelectrical impedance (Tanita, BC420SMA, Weda B.V., Naarden, Holland). BMI (kg/m$^2$) was calculated based on body height and body mass. Finally, waist circumference (0.1 cm) was measured using a flexible tape measure.

Motor imagery was tested with a classic hand laterality judgment task (HLT (*Parsons, 1987*)). Single-hand stimuli (9 by 8 cm) were presented on a laptop screen (Dell Precision M6700, 17-inch) using OpenSesame (version 3.0.7) (*Mathot, Schreij & Theeuwes, 2012*). The participant sat at a distance of approximately 60 cm from the screen and was instructed to indicate the laterality of the stimulus by pressing the keyboard (i.e., letter "d" or "k" on a qwerty keyboard for a left or right hand, respectively), while imagining that the hand on display was his/her own hand. The pictures of the hands were presented with palm facing up or down at an angle of 0°, 60°, 120°, 180°, 240°, or 300° (see Fig. 2). Before presentation of the stimulus a fixation cross was shown in the center of the screen, which was replaced by the actual hand stimulus after a random interval between 1,300 and 1,800 ms to avoid anticipatory responses. After a practice and familiarization period of five trials, during which it was ensured that the participants understood the instructions, three blocks of trials were recorded. Each block contained two trials per combination of stimuli [2 hands (left, right), 2 sides (palm, back), 6 orientations (0°−300° ); $N = 24$], giving a total of 144

**Table 1 Descriptive statistics (mean ± standard deviation) for anthropometric measurements and motor competence of the three groups.** Children with obesity + motor impairment (OB−), children with obesity without motor impairment (OB+) and children with healthy weight without motor impairment (HW). The final column reports the outcome of the ANOVA to explore between between-group differences.

| | OB−<br>$N = 19$ | OB+<br>$N = 13$ | HW<br>$N = 18$ | ANOVA<br>$F(2,47)$ |
|---|---|---|---|---|
| *Demographic characteristics* | | | | |
| Gender (boys/girls) | 6/13 | 7/6 | 14/4 | |
| Age | 9.9 ± 1.1 | 9.2 ± 1.0 | 9.5 ± 1.3 | 1.269 |
| *Anthropometric measurements* | | | | |
| Body height (cm) | 145.2 ± 7.8 | 141.9 ± 7.8 | 140.2 ± 8.9 | 1.739 |
| Body weight (kg) | 68.7 ± 18.3 | 59.9 ± 8.8 | 33.4 ± 5.7 | 37.993[*] |
| Body fat (%) | 44.8 ± 9.1 | 43.8 ± 5.9 | 16.8 ± 4.2 | 93.079[*] |
| Waist circumference (cm) | 94.7 ± 13.3 | 91.3 ± 7.5 | 61.1 ± 4.5 | 67.735[*] |
| Body mass index (kg/m$^2$) | 32.2 ± 6.5 | 29.7 ± 2.5 | 16.9 ± 1.2 | 64.891[*] |
| *Motor competence* | | | | |
| General motor competence | 46.8 ± 12.2 | 71.9 ± 10.1 | 82.4 ± 6.5 | 62.311[*] |
| Manual dexterity | 18.8 ± 8.1 | 27.3 ± 6.2 | 30.2 ± 4.3 | 15.497[*] |
| Ball skills | 14.4 ± 5.5 | 19.7 ± 5.0 | 21.1 ± 4.5 | 8.886[*] |
| Balance skills | 13.6 ± 5.5 | 24.9 ± 7.0 | 31.1 ± 3.3 | 51.394[*] |

**Notes.**

Children with obesity + motor impairment (OB−), children with obesity without motor impairment (OB+) and children with healthy weight without motor impairment (HW). The final column reports the outcome of the ANOVA to explore between between-group differences.

[*]$p \leq 0.001$.

trials per participant. Per trial, the software recorded the accuracy (correct or incorrect) and the response time (RsT) to the nearest ms.

## Analysis and statistics

After deletion of anticipatory responses (RsT < 250 ms) and late or absent responses (RsT ≥ 8,000 ms), mean RsTs of the remaining trials (correct and incorrect) were computed for each of the stimulus presentation conditions per individual. Note that the orientations of both hands were flipped such that angles between 0° and 180° represented medial rotations; angles between 180° and 360° represented lateral rotations. In addition to that, accuracy (ACC) was calculated as the proportion of correct responses at each of the stimulus presentation conditions per individual. As preliminary analysis indicated that there was a positive linear relationship between overall mean RsT and ACC, it was deemed appropriate to calculate the inverse efficiency score (IES), by dividing the RsT by the proportion of correct responses at each stimulus presentation (*Townsend & Ashby, 1978*; *Townsend & Ashby, 1983*). IES combines speed and error in one metric, yet it inflates variance disproportionally in cases were proportion correct is below chance (see *Bruyer & Brysbaert, 2011* for a detailed argumentation). The proportion correct in our data ranged from 0.48 to 0.97, therefore it was decided to calculate IES only for those subjects who had proportion correct scores above chance level. Based on a binomial distribution with

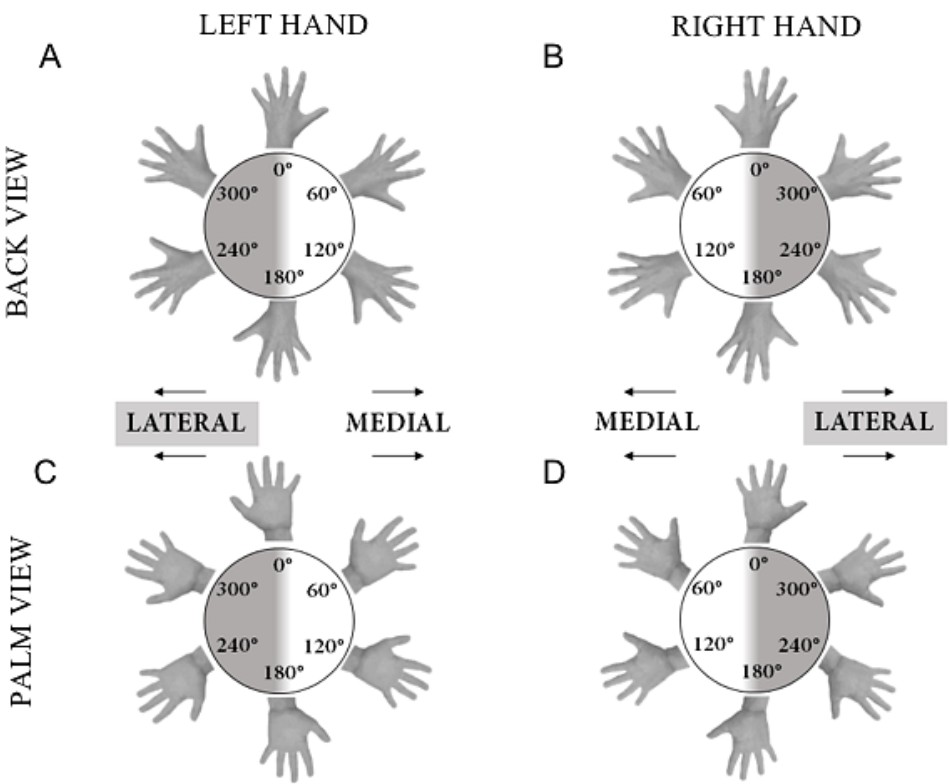

**Figure 2  Illustration of hand stimuli used in the hand laterality judgement task.** The stimuli are shown at each of the six possible angles of rotation. A, palm of the left hand; B, palm of the right hand; C, back of the left hand; D, back of the right hand.

$p = 0.50$ for each trial, individual performance was considered significantly above chance level when more than 76 out 144 trials (52.8%) were correct.

Between-group differences in anthropometric measurements and M-ABC2 scores were examined using separate univariate ANOVAs. To investigate group differences in motor imagery performance three separate repeated measures ANOVA were run for the dependent variables RsT, ACC and IES with Group (OB-, OB+, HW) as between groups factor and Hand (left, right), Side (palm, back), and Angle (0°, 60°, 120°, 180°, 240°, 300°) as within group factors. Within these analyses our first focus was on the effect of rotation, including potential interactions with the factor Group. Larger RsTs and smaller ACC for greater deviations from the normal orientation (i.e., with the fingers pointing upwards) were indicative of the use of mental rotation to judge laterality. Secondly, we looked at the difference between medial and lateral rotations and the effect of Side to examine whether the judgments obeyed to the anatomical and biomechanical constraints that act upon actual movements. Here, smaller RsTs for medial vs. lateral rotations and for hand back vs. hand palm were indicating the use of motor imagery. All analyses were run with IBM SPSS Statistics version 25. Effects with $p < 0.05$ were considered significant and partial eta squared values ($\eta^2$) were reported to indicate effect size where appropriate. To correct for multiple comparisons, Bonferroni adjustments were applied.

## RESULTS

### Individual characteristics

Descriptive statistics of the anthropometric measurements and motor competence scores are shown in Table 1. The ANOVAs revealed that children with obesity (OB- and OB+) were significantly heavier ($p \leq 0.001$) and had a higher percentage of body fat ($p \leq 0.001$), waist circumference ($p \leq 0.001$) and body mass index ($p \leq 0.001$) compared to HW controls. No significant differences in anthropometric measurements were observed between the OB+ and OB- group ($p > 0.05$). For general motor competence, significant between-group differences were observed, with the HW group performing better than the OB+ ($p = 0.016$) and OB- group ($p < 0.001$), and the OB+ group performing better than the OB- group ($p < 0.001$). A similar between-group difference was found for the sub-score on balance (HW>OB+, $p = 0.007$; HW>OB-, $p < 0.001$; OB+>OB-, $p < 0.001$). For manual dexterity and ball skills the performances of the HW and OB+ group did not differ, but both groups had significantly higher scores than the OB-group (manual dexterity: HW>OB-, $p < 0.001$; OB+>OB-, $p = 0.002$; ball skills: HW>OB-, $p = 0.001$; OB+>OB-, $p = 0.016$).

### Mental rotation and motor imagery performance

The ANOVAs indicated that the effect of Hand stimulus (right or left) was not significant and was not involved in any of the interaction effects for any of the dependent variables (RsT, ACC, and IES). Therefore, this independent factor was removed from further analyses.

For the RsT, we observed a significant effect of Side [$F(1,47) = 41.689$, $p < 0.001$, $\eta^2 = 0.470$], Angle [$F(5,235) = 35.225$, $p < 0.001$, $\eta^2 = 0.428$], and an interaction effect between these factors [Side x Angle $F(5,235) = 18.237$, $p < 0.001$, $\eta^2 = 0.280$]. Further investigation indicated that responses to stimuli of hand palms were generally slower than responses to those showing the back of the hand ($2,910 \pm 98$ ms vs. $2,543 \pm 88$ ms). For the effect of Angle, it was found that for both sides RsTs to medially rotated stimuli ($60°$ and $120°$) were smaller than RsTs to laterally rotated stimuli ($240°$ and $300°$), however, this effect was more pronounced for hand back than for palm of hand (see Figs. 3A and 3B). A main effect of Group or any interaction with this factor remained absent.

Proportion correct (ACC) was smaller in OB- group ($72.0 \pm 2.0\%$) compared with the OB+ ($85.7 \pm 3.6\%$) and HW group [$88.5 \pm 3.0\%$; main effect of Group: $F(2,47) = 7.525$, $p = 0.001$, $\eta^2 = 0.243$]. No difference was found for ACC between the HW and OB+ children. Furthermore, a main effect of Side [$F(1,47) = 19.623$, $p < 0.001$, $\eta^2 = 0.295$] and Angle [$F(5,235) = 15.224$, $p < 0.001$, $\eta^2 = 0.245$], as well as an interaction between these two factors was observed [$F(5,235) = 5.823$, $p < 0.001$, $\eta^2 = 0.110$]. Closer inspection of these effects revealed better ACC when the stimulus was rotated over $0°$, $60°$ or $120°$ vs. rotations over $180°$ or $240°$, with different profiles for palms and backs (see Figs. 3C and 3D). No interactions with Group were found.

Five participants, one of the HW and four of the OB- group, demonstrated ACC scores below chance (range: $47.9$–$52.1\%$). After removing the results of these participants, the ANOVA on the IES indicated a main effect of the factors Group [$F(2,40) = 3.384$, $p = 0.044$, $\eta^2 = 0.145$], Side [$F(1,40) = 13.410$, $p = 0.001$, $\eta^2 = 0.251$] and Angle [$F(5,200) = 16.266$,

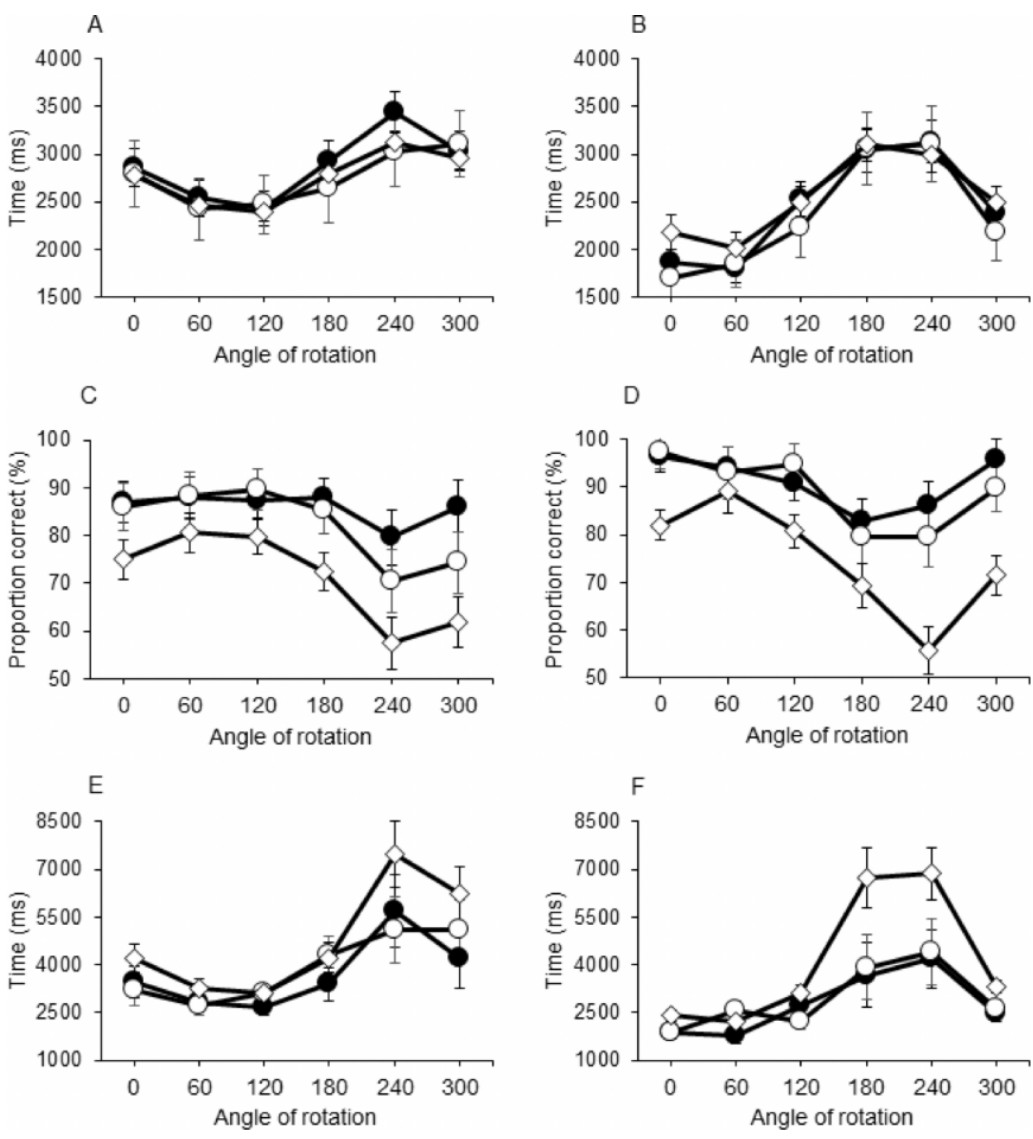

**Figure 3  Response Time (A & B), Proportion Correct (C & D) and Inverse Efficiency Score (E & F).** The group of children with obesity and motor impairments (OB−) is indicated with ◇; Children with obesity without motor impairments (OB+) are indicated with ◯: Healthy weight control group (HW) is indicated with ●. The stimulus on display was the palm of the hand in (A, C, E); the back of the hand is shown in (B, D, F).

$p < 0.001$, $\eta^2 = 0.289$]. Furthermore, there was a two-way interaction between Side and Angle [F(5,200) = 3.894, $p = 0.002$, $\eta^2 = 0.089$] and a three-way interaction between Group, Angle and Side [F(10,200) = 2.253, $p = 0.016$, $\eta^2 = 0.089$]. Post-hoc inspection showed that efficiency was generally better (i.e., IES smaller) for back of the hand vs. hand palm. In addition, the HW group had better IES than the OB- group ($p = 0.050$), in particular at angles of 0°, 240° and 300° for palms and 180°, 240° and 300° for backs. No significant difference was found between the OB- and the OB+ group, or between the OB+ group and the HW group (see Figs. 3E & 3F).

## DISCUSSION

This study set out to examine the capacity to enlist motor imagery during a mental rotation task in children with obesity. In addition to this, it was investigated whether performance on this task was related to the children's motor competence. To this end, the response time, accuracy and efficiency of the children with obesity with scores below the 5th percentile on the MABC-2 (OB-) was compared to those of the other children with obesity (OB+) and those with healthy weight (HW) using a classic hand laterality task. It was found that responses of children with obesity (OB- and OB+) were as fast as those in healthy weight control children (HW). Proportion of correct responses and efficiency were smaller in the group with obesity and motor difficulties (OB-), but the judgements were influenced by the side of the stimulus and its angle of rotation to a similar extent in all three groups.

Consistent with previous studies, stimuli with greater angular deviations resulted in slower responses, which indicates that mental rotation was used to judge the laterality of the hand on display. Indeed, the hand laterality task requires visual spatial cognition and reasoning, and it has been shown that the duration of these processes increases with angular disparity between the orientation of the stimulus and the "normal" orientation. There is evidence that this ability is related to problem solving (*Geary et al., 2000*) and the acquisition of mathematical knowledge (*Hegarty & Kozhevnikov, 1999*), and is likely to be involved in sport and movement skills (*Kaltner, Riecke & Jansen, 2014*; *Jansen, Lehmann & Doren, 2012*). In keeping with this, the children with obesity and motor difficulties in the present study made more errors and were less efficient than controls. Proportion correct and inverse efficiency scores of children with obesity without motor problems were not different from those in healthy weight children. This difference between the two groups with obesity suggests reduced spatial reasoning skills, but only in those with severe motor difficulties.

Another objective of this study was to investigate the embodied nature of mental rotation or motor imagery in hand laterality judgments in children with obesity with and without motor impairments. This is important as the ability to enlist motor imagery is linked to a person's internal action representation capacity, which is deemed to be essential for motor planning and control (*Desmurget & Grafton, 2000*). In all three groups response times were longer when the side of the hand on display was incongruent with the posture of the participant (longer response time for palm vs. back). In addition, stimuli rotated to the lateral side (300° and 240°) led to longer response times than stimuli rotated to the medial side 60° and 120°. In other words, the behavior of all three groups conformed to the anatomical and/or biomechanical constraints that act on actual hand rotations. This indicates that, irrespective of group, hand laterality judgements appear to be solved automatically using embodied mental spatial transformations of the viewer (1st person perspective) rather than of the viewed object (3rd person perspective). This is in contrast to what has been observed in other populations with motor impairments, where motor imagery is only used after specific instruction (*Williams et al., 2006*; *Williams et al., 2008*).

Our results corroborate earlier findings of Kaltner et al. (*Li et al., 2008*), who found that adolescents with obesity were able to engage in egocentric transformations, yet with slower

response times than their healthy weight peers for larger angular deviations. Despite the fact that the responses in this mental rotation task appear to be embodied, and involve motor imagery, the overall performance of children and adolescents with obesity is reduced, yet in the current study only in those with motor impairments. Preliminary evidence suggests that the poor performance in this task is associated with reduced excitability of the primary motor cortex when engaging in motor imagery (*Hyde et al., 2018*). It thus seems that the visual-spatial and action representation processes underlying motor imagery are affected in this group. Given the known relationship between impaired motor imagery and motor control issues (*Fuelscher et al., 2015*), our findings provide support to the notion that the motor impairment of these children is not just a matter of excessive weight but may be related to more fundamental deficits in motor control. Indeed, previous research signaled the potential relationship between DCD and obesity (*Cairney et al., 2005*; *Hendrix, Prins & Dekkers, 2014*), calling for special attention in the treatment of this group.

Clearly, the relationship between obesity and motor control problems is complex, and the present study does not allow to infer cause and effect. The association found between obesity, motor impairments and mental rotation deficits warrants further investigation. In this respect, it is also relevant to note that children with motor impairments often withdraw from movement opportunities (*Smyth & Anderson, 2000*). Limited movement activity may, in turn, have a negative impact on the development of spatial cognition, so the relationship between motor competence and mental rotation capacity is likely to be reciprocal (*Vasilyeva & Lourenco, 2012*). Furthermore, nutritional research in rats has shown that diet-induced obesity due to excess sucrose intake may lead to impaired spatial learning and long-term spatial memory (*Jurdak, Lichtenstein & Kanarek, 2008*). These findings highlight that obesity is a multifaceted problem that requires a multidisciplinary approach with attention for dietary and lifestyle factors, as well as cognitive and motor competence (*Liang et al., 2014*; *Simmonds et al., 2016*).

By contrasting the mental rotation performance of children with obesity with and without motor impairments, the current study has provided an extra dimension to earlier findings of Jansen and colleagues (*Jansen et al., 2011*; *Kaltner, Schulz & Jansen, 2017*). Here, we have shown that deficits in mental rotation are a particular problem when obesity is accompanied by motor impairments, however, a number of limitations need to be considered. First, the design lacks a group of HW children with motor impairments. This would have enabled a direct assessment of the variance related to the relevant factors body weight/obesity and motor impairment and the interaction between the two. Secondly, the children with obesity were recruited from a specialized rehabilitation center, meaning that sampling was not fully randomized. In fact, the children are referred to this center by their general practitioner or pediatrician based on the severity of their weight problem and failure of conventional care. Our findings may therefore pertain to the more severe cases of obesity. Furthermore, data on the intellectual capacity or processing speed of our sample were not available. Given that spatial cognition is known to correlate with mathematical capacity it would have been desirable to control for this factor. In this respect, it is also important to note that five out of 19 children of the OB- group attended a school for special education. While these children were free from neurological conditions, (subtle)

neuropsychological issues cannot be excluded. As for processing speed, we know that children with obesity demonstrate increased reaction times in a simple reaction time task (*Gentier et al., 2013a*). Whether or not the current findings reflect a specific issue with spatial processing and action representation or rather a general slowness, should be the subject of future research. Finally, our results only provide insight into one specific aspect of the central processes related to spatial cognition and motor control, i.e., mental rotation capacity and motor imagery. To unravel the motor impairments of children with obesity further, more research is needed.

## CONCLUSIONS

The results of this study demonstrate that children with obesity engage in motor imagery during hand laterality judgments in this mental rotation tasks, regardless of the fact that they have motor impairments. The performance of children with obesity and motor impairments is, however, flawed (i.e., slower and less efficient) compared to their counterparts without motor impairments and children with a healthy weight. It thus seems that the visual-spatial and action representation processes underlying motor imagery are affected in this group. A deficit in these processes lends support to the notion that the motor impairments are related to more fundamental deficits in motor control and are not just a matter of excessive weight. Due to the specific nature of the sample and the lack of potential contributing factors, e.g., intellectual capacity, our findings should be treated with caution, however, for practitioners it is important to acknowledge the potential presence of motor impairments in children with obesity.

## ACKNOWLEDGEMENTS

The authors are very grateful to all participants and their parents, the staff from the rehabilitation center "Zeepreventorium" (De Haan, Belgium) and the board of the participating schools. All authors conceived and designed the experiment.

### Funding
The study was funded by the PhD fellowship of the Flemish Research Council (FWO) awarded to Mireille Augustijn [3F000714]. The funders had no role in study design, data collection and analysis, decision to publish, or preparation of the manuscript.

### Grant Disclosures
The following grant information was disclosed by the authors:
PhD fellowship of the Flemish Research Council (FWO): 3F000714.

### Competing Interests
The authors declare there are no competing interests.

## Author Contributions

- Frederik J.A. Deconinck conceived and designed the experiments, analyzed the data, contributed reagents/materials/analysis tools, authored or reviewed drafts of the paper, approved the final draft.
- Eva D'Hondt and Karen Caeyenberghs conceived and designed the experiments, authored or reviewed drafts of the paper, approved the final draft.
- Matthieu Lenoir conceived and designed the experiments, contributed reagents/materials/analysis tools, authored or reviewed drafts of the paper, approved the final draft.
- Mireille J.C.M. Augustijn conceived and designed the experiments, performed the experiments, analyzed the data, contributed reagents/materials/analysis tools, prepared figures and/or tables, authored or reviewed drafts of the paper, approved the final draft.

## Human Ethics

The following information was supplied relating to ethical approvals (i.e., approving body and any reference numbers):

The Ethics Committee of the Faculty of Medicine and Health Sciences of Ghent University granted ethical approval for this study (Ethical application Ref: 2014/0003).

## Data Availability

The raw measurements and participants characteristics are available in the Supplemental Files.

## Supplemental Information

Supplemental information for this article can be found online at http://dx.doi.org/10.7717/peerj.8150#supplemental-information.

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
