# Peer review of "The association between mental rotation capacity and motor impairment in children with obesity—an exploratory study"

_PeerJ, doi:10.7717/peerj.8150_

## Round 0.1 · original submission · Major Revisions

While the reviewers generally commented positively, they also raised some methodological concerns. I would like the authors to provide a stronger (theoretical) rationale for the statistical analyses that were performed but would advise against adding hypotheses after the results are known. The authors should also correct for multiple comparisons, including the multiple correlations reported in Table 2.

Reviewer 1 ·

Basic reporting

no comment

Experimental design

no comment

Validity of the findings

no comment

Additional comments

Review:
Mental rotation capacity in children with obesity is reduced, but only those with motor imairments


The authors present a mental rotation study with children with obesity and motor impairments (OB-), with children without motor impairments (OB+) and with children with healthy weight (HW). The children were between 7-11 years old and they had to solve a mental rotation task with hands as stimuli and a motor task (M-ABC). The results show that the OB- group had significantly lower accuracy rates and inverse efficiency scores than the HW group. There was no difference between the HW group and the OB+ group. Due to this, the authors assume that the impaired motor impairment relates to the worse mental rotation performance in children with obesity.

In my opinion, the study is interesting and worth to be published in PeerJ. However, my major concern is the (limited) strength of the theoretical rationale and also the missing data of a cognitive processing speed measurement. For this, I suggest a major revision of the paper.


Introduction
• The theoretical rationale could be ameliorated. The distinction of egocentric vs. object-based mental rotation task is missing and also the embodiment approach in mental rotation research.
• Line 47/48:This focus contradicts the title of the study.
• The following important paper is missing: Kaltner, S., Schulz, A., & Jansen, P. (2017). The association between obesity and mental rotation ability in an adolescent sample. Obesity Research & Clinical Practice, 11, 127-129.
• Can the authors formulate direct hypotheses?

Methods &Materials
• Are the children in the HW and obesity group matched by age and sex?
• Why did the authors present the stimuli in a random time interval?
• Line 164: “obesity” is missing
• 7 out of 25 in the HW group is a high number. What about the mental rotation performance of those children?
• Line 191: Why did the authors choose these time windows for quick and late responses?
• The calculation of the IES is rather unusual.
• Did the authors correct for non-sphericity of the data?

Results
• Line 240/241: p-values are missing.
• Line 274: Is this significant after correction for multiple testing?

Discussion
• The discussion is well written but I am missing the consolidation of the theoretical rationale.
• There are some studies with children with Spina Bifida showing an impaired mental rotation performance. Those studies might be relevant.
• The authors mention that cognitive processing speed was not measured. This is an important point especially because they emphasize the importance of motor planning processes. The authors should elaborate on this point.

Reviewer 2 ·

Basic reporting

Some minor English editing required.
Sufficient literature incorporated, however a few references need to be added to support some statements.
Professional article structure - YES.

Some results do not appear to be linked to the aims of the study.

Experimental design

Aims of study need to be better aligned to the results presented or the results should be amended to support the original stated aims. Specific notes provided in the attached document.

The building of the rationale in the introduction is not supported by the methodological approach used. Please see comments regarding the need to add an additional group (that you have currently excluded for analysis).

Validity of the findings

Statement of main findings are flawed by methods of analysis and decision to exclude a required group for comparison (Healthy weight children with impaired motor skills).

Additional comments

Thank you for the opportunity to review ‘Mental rotation capacity in children with obesity is reduced, but only those with motor impairments’. This is an interesting paper using quantitative methods to explore the mental rotation capacity in children with obesity, and specifically their ability to use motor imagery as a proxy of motor planning and control.

I have provided a separate document with line by line comments for your consideration. Should you be able to make these major amendments I would encourage you to resubmit a revised version of the manuscript.

Annotated reviews are not available for download in order to protect the identity of reviewers who chose to remain anonymous.

---

## Round 0.2 · Minor Revisions

While the revised manuscript has improved considerably, there are a few outstanding comments that need to be addressed before the manuscript can be accepted for publication. In particular, the authors should clarify whether their study is exploratory in nature and interpret their findings accordingly.

Reviewer 1 ·

Basic reporting

Please see comments to the authors

Experimental design

Please see comments to the authors

Validity of the findings

Please see comments to the authors

Additional comments

Review revision:
Mental rotation capacity with obesity is reduced but only in those in motor impairments


The authors revised their manuscript in some points according to my suggestions. However, I have some concerns, so I can not recommend publication of this manuscript as it stands.


1. First of all, the authors should not formulate post hoc hypotheses according to their data but in my opinion it is necessary to provide a theoretical rationale for the formulation of hypotheses BEFORE gathering the data. Because the authors have not done this, their research seemed not to be theoretical driven, which have to doubt the worthness of publishing this study in a journal like PEERJ.
2. In my opinion, the differentiation in egocentric and object-based transformation due to the stimuli used is not correct, please consider the work of Voyer, Jansen, & Kaltner, 2017 in QJEP.
3. The post-hoc analysis of the 7 HW children is not included in the manuscript. I also agree with reviewer 2 regarding this point. This point should be regarded seriously.

Reviewer 2 ·

Basic reporting

Please see comments in attached document from Reviewer 2.

Experimental design

Please see comments in attached document from Reviewer 2.

Validity of the findings

Please see comments in attached document from Reviewer 2.

Additional comments

Overall, the majority of issues have been addressed by the authors. A few concerns remain regarding some responses and these have been added to the attached document. Please see comments in attached document from Reviewer 2. Should the authors be able to address these outstanding concerns I would support this manuscript being published. I would like to thank the authors for their efforts to address previous comments and thank them for their important contributions to this field.

Annotated reviews are not available for download in order to protect the identity of reviewers who chose to remain anonymous.

---

## Round 0.3 · Minor Revisions

PeerJ does publish exploratory studies. While it is not required to explicitly state in the title that the study is exploratory, I agree with the reviewer that the current title is too conclusive. I would like to ask the authors to reformulate the title and the conclusion section of the abstract in line with the more exploratory nature of the study.

Reviewer 1 ·

Basic reporting

I appreciate the explanation of the authors.
Please integrate: An explorative study in the title.

Experimental design

fine

Validity of the findings

fine

---

## Round 0.4 · accepted · Accept

The authors adequately addressed the outstanding comments.